

# Effect of vitamin D supplementation in patients with chronic hepatitis C after direct-acting antiviral treatment: a randomized, double-blind, placebo-controlled trial

Supachaya Sriphoosanaphan[1,2], Kessarin Thanapirom[1,2,3], Stephen J. Kerr[4], Sirinporn Suksawatamnuay[1,2,3], Panarat Thaimai[1], Sukanya Sittisomwong[1], Kanokwan Sonsiri[1], Nunthiya Srisoonthorn[2], Nicha Teeratorn[1], Natthaporn Tanpowpong[5], Bundit Chaopathomkul[5], Sombat Treeprasertsuk[1], Yong Poovorawan[6] and Piyawat Komolmit[1,2,3]

[1] Division of Gastroenterology, Department of Medicine, Faculty of Medicine, Chulalongkorn University, Bangkok, Thailand
[2] Center of Excellence in Liver Diseases, Thai Red Cross, King Chulalongkorn Memorial Hospital, Bangkok, Thailand
[3] Liver Fibrosis and Cirrhosis Research Unit, Chulalongkorn University, Bangkok, Thailand
[4] Biostatistics Excellence Center, Department of Research Affairs, Chulalongkorn University, Bangkok, Thailand
[5] Department of Radiology, Faculty of Medicine, Chulalongkorn University, Bangkok, Thailand
[6] Center of Excellence in Clinical Virology, Faculty of Medicine, Chulalongkorn University, Bangkok, Thailand

Corresponding author
Piyawat Komolmit,
pkomolmit@yahoo.co.uk

## ABSTRACT

**Background**. Replacement of vitamin D (VD) among patients with chronic hepatitis C (CHC) before viral eradication has demonstrated a protective effect on serum markers associated with hepatic fibrogenesis. We therefore hypothesized that VD may facilitate further fibrosis amelioration following curative treatment with direct-acting antivirals (DAA).

**Methods**. This study was a randomized, double-blind, placebo-controlled trial conducted between February 2018 and August 2018. Patients with CHC and VD deficiency were randomized in a 1:1 ratio to either receive ergicalciferol or placebo over 6 weeks. Biochemical analysis indicators, including 25-hydroxyvitamin D (25(OH)D), fibrogenic markers [(transforming growth factor beta 1 (TGF-$\beta$1) and tissue inhibitors of matrix metalloproteinases 1 (TIMP-1)], and fibrolytic markers [matrix metalloproteinase 9 (MMP-9) and amino terminal type III procollagen peptide (P3NP)], were assessed at baseline and at 6 weeks. Serum 25(OH)D was analyzed by a chemiluminescence immunoassay. Serum hepatic fibrogenesis markers were measured using a quantitative sandwich enzyme-linked immunosorbent assay.

**Results**. Seventy-five patients with CHC and VD deficiency were randomly assigned to VD ($n = 37$) and placebo ($n = 38$) groups. At the end of the study, the mean serum 25(OH)D level had risen to a normal level in the VD group, but was still deficient in the placebo group (41.8 $\pm$ 9.1 vs. 18.1 $\pm$ 4.6 ng/mL, $p < 0.001$). Upon restoration of the VD level, there were no significant mean differences in the change from baseline for TGF-$\beta$1 (−0.6 ng/mL (95% confidence interval (95% CI) [−2.8–1.7]), $p = 0.63$),

TIMP-1 ($-5.5$ ng/mL (95% CI [$-26.4$ –$15.3$]), $p = 0.60$), MMP-9 (122.9 ng/mL (95% CI [$-69.0$ –$314.8$]), $p = 0.21$), and P3NP ($-0.1$ ng/mL (95% CI [$-2.4$ –$2.2$]), $p = 0.92$) between the VD and placebo groups.

**Conclusion**. Short-term VD supplementation after DAA treatment in patients with CHC does not improve serum fibrogenesis markers and may not expedite the residual liver fibrosis healing process. Future studies are warranted to evaluate the long-term effect of VD supplementation on hepatic fibrosis regression.

## INTRODUCTION

Hepatic fibrogenesis is a complex, dynamic cellular process regulated by a balance between fibrogenic and fibrolytic activities. Liver fibrogenic cytokines, such as transforming growth factor beta 1 (TGF-$\beta$1) and tissue inhibitor of matrix metalloproteinase-1 (TIMP-1), initially activate the quiescent hepatic stellate cells (HSCs) after liver injury. This process consequently leads to a change in the fibrotic cascade and results in an excessive accumulation of extracellular matrix (ECM) within the liver parenchyma (*Lee, Wallace & Friedman, 2015*). Subsequently, the cellular healing process eventually occurs after the resolution of the injury. Matrix metalloproteinases (MMPs), which are important fibrolytic enzymes, play a pivotal role in matrix remodeling during the recovery phase. MMPs regulate collagen degradation, thereby facilitating liver fibrosis regression (*Okazaki et al., 2000*).

Hepatitis C virus (HCV) infection remains a major health-care problem worldwide, leading to high morbidity and mortality from cirrhosis and hepatocellular carcinoma (HCC) (*Stanaway et al., 2016*). Notably, the advent of direct-acting antiviral (DAA) therapy for HCV has transformed clinical practice and provided >90% cure rate among patients with chronic hepatitis C (CHC) (*European Association for the Study of the Liver, 2018*). Amelioration of liver inflammation and improvement of hepatic synthetic function have been demonstrated during the course of treatment and shortly after the completion of DAA therapy (*Charlton et al , 2015*; *Curry et al., 2015*). Additionally, a remarkable reduction in liver stiffness has also been shown after successful HCV eradication (*Dolmazashvili et al., 2017*; *Pan et al., 2018*). However, a large number of patients with CHC still have substantial residual liver fibrosis after the viral cure, thereby remaining as an important concern in current clinical practice (*Giannini et al., 2019*; *Singh et al., 2018*). Recent studies have demonstrated that patients with CHC who still have advanced liver fibrosis and cirrhosis after a sustained virological response (SVR) remain at high risk of HCC (*Su & Ioannou, 2019*). Thus, guidelines recommend continuing an HCC surveillance program for such patients (*European Association for the Study of the Liver, 2018*; *A-IHG Panel, 2018*).

Vitamin D (VD) deficiency/insufficiency is a global problem affecting approximately 1 billion people; its prevalence varies by age and ethnicity in different countries (*Amrein et*

*al., 2020*). A number of studies have reported a high prevalence of VD deficiency in patients with chronic liver diseases, regardless of etiology (*Arteh, Narra & Nair, 2010*; *Malham et al., 2011*). Notably, >60% of patients with CHC experience VD deficiency (*Jin, Chen & Sheng, 2018*). Low serum VD levels could lead to negative liver-related consequences and liver fibrosis progression (*Dadabhai et al., 2017*; *Kubesch et al., 2018*). Moreover, in a recent clinical study, since the liver is an intermediate organ of VD metabolism, a high proportion of patients with CHC still have VD deficiency and insufficiency after successful HCV eradication (*Backstedt et al., 2017*; *Sriphoosanaphan et al., 2020*).

Growing experimental and clinical evidence has revealed an anti-fibrogenic effect of VD (*Baur et al., 2012*; *Ding et al., 2013*; *Wahsh et al., 2016*). Our previous study demonstrated a considerable improvement in serum fibrogenesis markers after VD restoration in patients with CHC before HCV treatment (*Komolmit et al., 2017*). This improvement could be due to the influence of VD on HCV replication, inflammation reduction, or the direct effect of VD on hepatic fibrogenesis. In this study, we aimed to further clarify the exact role of VD in hepatic fibrogenesis amelioration in patients with CHC who underwent curative treatment with DAA.

## MATERIALS & METHODS

### Study population and study design

This randomized, double-blind, placebo-controlled study was conducted at King Chulalongkorn Memorial Hospital, which is a tertiary referral center and academic teaching hospital, in Bangkok, Thailand, from February 2018 to August 2018. All patients with CHC who achieved SVR within 1 year after DAA therapy in the hepatology clinic were assessed for eligibility. The inclusion criteria were as follows: age between 18 and 80 years, VD insufficiency/deficiency (defined as serum 25(OH)D level <30 ng/mL), and evidence of more than F2 liver fibrosis from either Fibroscan® (transient elastogram ≥7.1 kPa (*Castera et al., 2005*), ultrasound elastography (≥6.7 kPa by 2D shear-wave elastography (*Bende et al., 2017*), or magnetic resonance elastography (≥3.2 kPa by two-dimensional gradient recall echo (*Ichikawa et al., 2012*)). Patients with a recent history of VD supplementation, evidence of chronic hepatitis B or human immunodeficiency virus infection, decompensated liver cirrhosis, autoimmune diseases, hepatocellular carcinoma, active infections from other pathogens, a history of steroid or immunosuppressive agent use, or a history of interferon treatment within 12 months, and those who were pregnant or lactating, were excluded.

Data were collected as previously described in *Komolmit et al. (2017)*. Specifically, randomization was performed with a computer-generated allocation sequence. The eligible patients were randomly assigned to receive VD or placebo (1:1 allocation using a random block size of 4). The process was performed by a research assistant who was not involved in the study. Randomization codes were secured in sealed envelopes until all data entry was complete.

**Table 1  Protocol for VD supplementation.**

| Diagnosis | 25(OH)D level (ng/mL) | Replacement (IU/week) | Ergocalciferol 20,000 IU/tab | Duration |
|---|---|---|---|---|
| Insufficiency | 20–30 | 60,000 | 2 tabs Monday and 1 tab Friday | 6 weeks |
| Deficiency | 10–20 | 80,000 | 2 tabs Monday and 2 tabs Friday | 6 weeks |
| Severe deficiency | <10 | 100,000 | 3 tabs Monday and 2 tabs Friday | 6 weeks |

Notes.
*VD, vitamin D; 25(OH)D, 25-hydroxyvitamin D; IU, international unit.

## Intervention

Vitamin D2 (VD2, ergocalciferol) and the placebo were prepared in identical capsules of the same weight by the Department of Pharmacy at King Chulalongkorn Memorial Hospital. All investigators and patients were blinded to the type of medication used until the end of the study. Each capsule of VD2 contained 20,000 international units of ergocalciferol. The protocol for VD supplementation was based on the 25(OH)D level at baseline (Table 1), which has been previously demonstrated to effectively increase VD levels in patients with CHC within a 6-week period (Komolmit et al., 2017). All patients initially received ergocalciferol or placebo on the day of randomization.

Baseline clinical characteristics, HCV genotypes, liver enzymes, and fibrosis-4 (FIB-4) and aspartate aminotransferase to platelet ratio index (APRI) scores were assessed. Blood samples were collected at two time points (baseline and at the end of the 6-week follow-up) and were kept at $-70\,°C$ until analysis. All blood samples were simultaneously analyzed for TGF-$\beta$1, TIMP-1, MMP-9, and amino terminal type III procollagen peptide (P3NP) levels at the end of the study. Pill counts and patient interviews were used to assess adherence to the prescribed medications. All participants were asked to refrain from outside-of-trial VD or multivitamin supplements and to maintain their normal activity. At the end of the study, patients who remained in a VD-insufficient state received VD supplementation as a standard of care.

## Outcomes and assessments

The primary outcome was the effect of VD supplementation on serum markers associated with hepatic fibrogenesis, including TGF-$\beta$1, TIMP-1, MMP-9, and P3NP, relative to the effect in the placebo group at the 6-week follow-up.

## Laboratory assessment
### Measurement of 25(OH)D level

Serum 25(OH)D levels, a functional indicator of VD status, were measured using the Liaison 25(OH) vitamin D total assay (DiaSorin, Saluggia, Italy); the measurement was performed on the LIAISON® chemiluminescence analyzer following the manufacturer's instructions. The final level is reported in ng/mL. According to the Endocrine Society Practice Guidelines (Holick et al., 2011), VD insufficiency/deficiency was defined as a 25(OH)D level <30 ng/mL. Specifically, serum 25(OH)D levels <20 ng/mL and between 20–29 ng/mL were defined as VD deficiency and VD insufficiency, respectively.

### Measurement of TGF-β1 level

The serum levels of human activated TGF-β1 were measured with a quantitative sandwich enzyme-linked immunosorbent assay (ELISA) technique following the manufacturer's instructions (Quantikine® ELISA, R&D Systems, Minneapolis, MN, USA). Before the assay, the latent TGF-β1 contained in the patients' serum was activated to the immunoreactive form using acid activation and neutralization. The results were calculated by reference to the standard curve.

### Measurement of TIMP-1 level

The serum levels of human TIMP-1 were measured using the quantitative sandwich ELISA technique according to the manufacturer's instructions (Quantikine® ELISA). The results were calculated by reference to the standard curve.

### Measurement of MMP-9 level

The serum levels of human MMP-9 protein were measured by a quantitative sandwich ELISA according to the manufacturer's instructions (Quantikine® ELISA). The results were calculated by reference to the standard curve.

### Measurement of P3NP level

The serum levels of P3NP were measured via a quantitative sandwich ELISA technique according to the manufacturer's instructions (Cloud-Clone Corp., TX, USA). The results were calculated by reference to the standard curve.

## Sample size calculation

Sample size calculations were based on data from our previous study on patients with CHC (*Komolmit et al., 2017*). Of the cytokines measured in both the present study and our previous study, we used the data of TGF-β1 for the calculation, since it resulted in the largest sample size. The mean difference in the change in TGF-β1 level from baseline to week 6 in the VD supplementation arm versus the placebo arm as a control was 81 pg/mL, with a pooled standard deviation of 110. Thus, enrolling 30 patients per group would give 80% power to detect a difference in TGF-β1 change from baseline between treatment groups of this size or more, at a 2-sided significance level of 5%. We inflated the sample size to 37 per randomized arm to account for possible dropouts.

## Statistical analysis

Baseline characteristics, including clinical and laboratory data, are presented as percentage or mean ± standard deviation (SD). In the initial analyses, the baseline characteristics of the patients were compared using t-tests or chi-square tests. Categorical variables were analyzed using the chi-square test or Fisher's exact test, as appropriate. For the primary study outcome of fibrogenesis marker changes, we used the independent $t$-test to evaluate mean differences and 95% confidence interval (95% CI) in the change from baseline to week 6 between the VD and the placebo groups. As a sensitivity analysis, to ensure that outliers did not exert an undue influence on the obtained between-arm estimates, we summarized the data as the median and 25th and 75th percentile levels, and used quantile regression

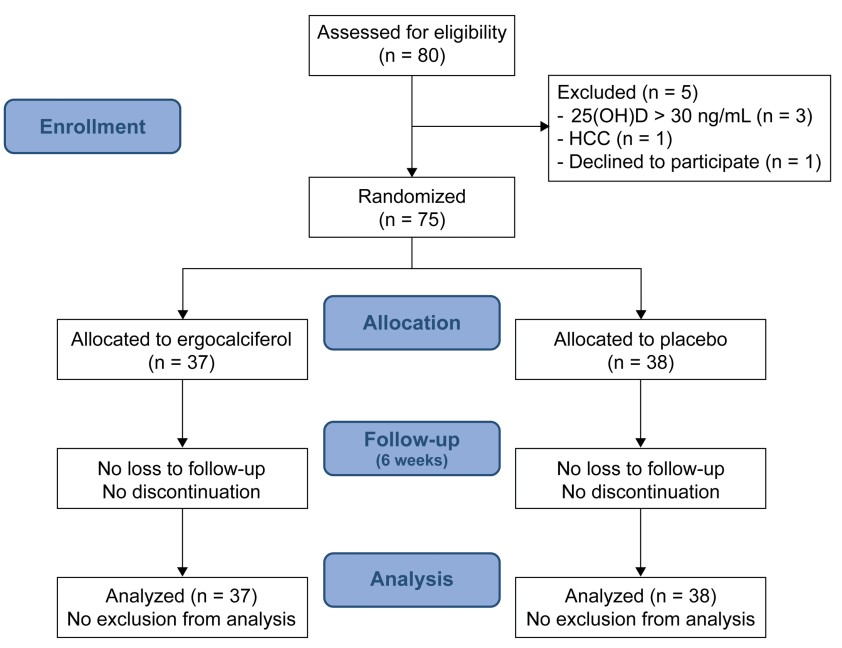

**Figure 1  Screening, randomization, and follow-up.**

to compare the median serum level changes between randomized arms. Statistical analyses were performed with Stata 16.1 (Statacorp, College Station, TX, USA).

## Ethical approval

This study was reviewed and approved by the Ethics Committee and Institutional Review Board (IRB) at Chulalongkorn University, Bangkok, Thailand, and was performed in accordance with the Declaration of Helsinki (1989) of the World Medical Association (IRB number: 707/60). Moreover, this trial was registered with the Thai Clinical Trials Registry (TCTR) based on the World Health Organization criteria on 22 November 2017 (TCTR20171206003). Prior to participation, all patients enrolled in this study provided written informed consent for participation, as well as for publication.

## RESULTS

### Patient characteristics

Between February 2018 and August 2018, 80 patients with CHC who achieved SVR after DAA treatment were screened for eligibility. Five patients were excluded from the study (Fig. 1). A total of 75 patients were finally included and randomly assigned to the VD group ($n = 37$) or the placebo group ($n = 38$). Mean age was 60.4 ± 7.8 years, patients were mostly female ($n = 45$, 60%), and a majority of the participants had cirrhosis ($n = 45$, 60%). As shown in Table 2, baseline characteristics between the groups were well balanced. No significant difference in any of the biochemical parameters and degree of liver fibrosis, as measured by FIB-4 and APRI scores, was found between the groups.

**Table 2  Baseline characteristics.**

|  | VD group (n = 37) | Placebo (n = 38) |
|---|---|---|
| Age (years) | 60.9 ± 7.8 | 59.4 ± 8.0 |
| Sex, male | 16 (43.2%) | 14 (36.8%) |
| Body mass index (kg/m$^2$) | 25.3 ± 4.3 | 24.3 ± 4.1 |
| HCV viral load before treatment (log IU/mL) | 5.9 ± 0.9 | 5.5 ± 1.4 |
| Genotype (n) | | |
| 1 | 19 | 18 |
| 3 | 17 | 17 |
| 6 | 1 | 3 |
| Cirrhosis | 21 (56.8%) | 24 (63.2%) |
| AST (U/L) | 29.6 ± 16.0 | 28.9 ± 10.9 |
| ALT (U/L) | 23.5 ± 11.0 | 23.7 ± 10.0 |
| DAA regimen | | |
| SOF/DAC | 11 | 4 |
| SOF/LED | 1 | 3 |
| SOF/VEL | 0 | 0 |
| SOF/DAC/RBV | 23 | 23 |
| SOF/LED/RBV | 1 | 7 |
| SOF/VEL/RBV | 1 | 1 |
| Fibrosis 4 score (FIB-4) | 2.8 ± 2.1 | 3.4 ± 2.8 |
| AST to platelet ratio index (APRI) | 0.6 ± 0.5 | 0.7 ± 0.6 |
| 25(OH)D level (ng/mL) | 17.2 ± 4.8 | 16.6 ± 4.1 |
| TGF-$\beta$1 level (ng/mL) | 16.3 ± 6.6 | 14.3 ± 8.3 |
| TIMP-1 level (ng/mL) | 236.9 ± 50.0 | 231.9 ± 50.3 |
| MMP-9 level (ng/mL) | 576.2 ± 296.2 | 570.1 ± 477.7 |
| P3NP level (ng/mL) | 28.6 ± 5.7 | 28.0 ± 4.3 |

**Notes.**

*VD, vitamin D; HCV, Hepatitis C virus; IU, international unit; AST, aspartate aminotransferase; ALT, alanine aminotransferase; DAA, direct-acting antiviral; SOF, sofosbuvir; DAC, daclatasvir; LED: ledipasvir; VEL, velpatasvir; RBV, ribavirin; 25(OH)D, 25-hydroxyvitamin D; TGF-$\beta$1, transforming growth factor beta 1; TIMP-1, tissue inhibitor of matrix metalloproteinase-1; MMP-9, matrix metalloproteinase 9; P3NP, amino terminal type III procollagen peptide.

## Changes in VD levels

Mean baseline serum 25(OH)D level was 17.2 ± 4.8 ng/mL and 16.6 ± 4.1 ng/mL in the VD and placebo groups, respectively. At the 6-week follow-up, the mean 25(OH)D level in the VD group increased to a normal level (41.8 ± 9.1 ng/mL), whereas a slight increase was observed in the placebo group (18.1 ± 4.6 ng/mL) (Fig. 2). The mean difference in 25(OH)D change from baseline to week 6 reflected a significantly greater change in the VD group than in the placebo group (23.1 ng/mL (95% CI [19.7–26.4]), $p < 0.001$).

Of note, three patients (8.1%) in the VD group still had a serum 25(OH)D level <30 ng/mL at 6 weeks after VD replacement; the 25(OH)D levels in two patients changed from an initial deficient state to an insufficient state after VD supplementation (absolute VD increase of 18.3 and 18.4 ng/mL, respectively), while the remaining patient had a slightly increased VD level (from 21.8 ng/mL at baseline to 23.5 ng/mL) after VD replacement.

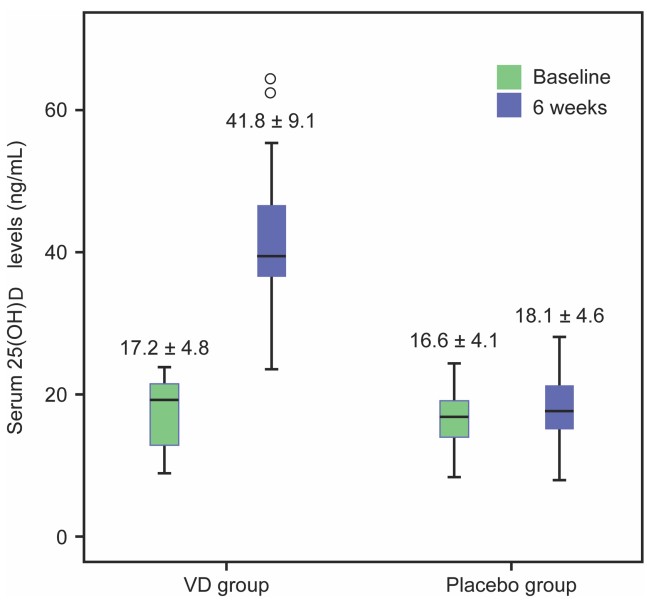

**Figure 2  Box plot of serum 25(OH)D at baseline and after VD or placebo supplementation for 6 weeks.**

All participants in the placebo group still had VD insufficiency/deficiency. All unused pills were returned at the last follow-up, and no patient reported missing any medications.

## Changes in serum hepatic fibrogenesis markers

Mean and median differences in serum marker changes from baseline to 6 weeks are shown in Tables 3 and 4, respectively, and in Fig. 3. Serum TGF-$\beta$1 levels slightly increased from 16.3 $\pm$ 6.6 ng/mL at baseline to 16.8 $\pm$ 7.6 ng/mL at 6 weeks in the VD group, and increased from 14.3 $\pm$ 8.3 to 15.4 $\pm$ 6.8 ng/mL in the placebo group. However, there was no significant difference in the mean change from baseline between the VD and placebo groups ($-0.6$ ng/mL (95% CI [$-2.8$ $-1.7$]), $p = 0.63$).

Similarly, mean serum TIMP-1 levels in the VD group slightly increased from 236.9 $\pm$ 50.0 ng/mL at baseline to 244.0 $\pm$ 66.0 ng/mL at 6 weeks. In the placebo group, mean serum TIMP-1 levels slightly increased from 231.9 $\pm$ 50.3 ng/mL at baseline to 244.5 $\pm$ 63.6 ng/mL at 6 weeks. There was no significant difference in the mean TIMP-1 change from baseline to 6 weeks between the VD and placebo groups ($-5.5$ ng/mL (95% CI [$-26.4$ $-15.3$]), $p = 0.60$).

Regarding fibrolytic markers, mean MMP-9 levels showed an increasing trend in the VD group (576.2 $\pm$ 296.2 ng/mL at baseline and 684.3 $\pm$ 423.5 ng/mL at 6 weeks); median levels showed a consistent pattern. In contrast, mean MMP-9 levels in the placebo group were approximately the same at baseline and week 6, but the median levels increased. There was no significant difference in the mean change from baseline at week 6 between the VD and placebo groups (122.9 ng/mL (95% CI [$-69.0$–314.8]), $p = 0.21$).

**Table 3** Serum biochemical markers levels at baseline and 6 weeks of follow-up, and the mean difference (95% CI) in serum marker change from baseline to 6 weeks in the vitamin D (VD) versus the placebo arm as a reference.

| Serum marker | Week | VD (n = 37) | Placebo (n = 38) | Mean difference (95% CI) in change from baseline to 6 weeks in VD vs placebo arm | p value |
|---|---|---|---|---|---|
| 25(OH)D level (ng/mL) | 0 | 17.2 ± 4.8 | 16.6 ± 4.1 | | |
| | 6 | 41.8 ± 9.1 | 18.1 ± 4.6 | 23.1 (19.7–26.4) | <0.001 |
| AST (U/L) | 0 | 29.6 ± 16.0 | 28.9 ± 10.9 | | |
| | 6 | 29.9 ± 15.6 | 30.0 ± 12.5 | −0.7 (−4.2–2.8) | 0.69 |
| ALT (U/L) | 0 | 23.5 ± 11.0 | 23.7 ± 10.0 | | |
| | 6 | 23.8 ± 11.1 | 27.1 ± 14.2 | −3.0 (−7.5–15) | 0.20 |
| TGF-$\beta$1 level (ng/mL) | 0 | 16.3 ± 6.6 | 14.3 ± 8.3 | | |
| | 6 | 16.8 ± 7.6 | 15.4 ± 6.8 | −0.6 (−2.8–1.7) | 0.63 |
| TIMP-1 level (ng/mL) | 0 | 236.9 ± 50.0 | 231.9 ± 50.3 | | |
| | 6 | 244.0 ± 66.0 | 244.5 ± 63.6 | −5.5 (−26.4–15.3) | 0.60 |
| MMP-9 level (ng/mL) | 0 | 576.2 ± 296.2 | 570.1 ± 477.7 | | |
| | 6 | 684.3 ± 423.5 | 555.2 ± 290.5 | 122.9 (−69.0–314.8) | 0.21 |
| P3NP level (ng/mL) | 0 | 28.6 ± 5.7 | 28.0 ± 4.3 | | |
| | 6 | 27.6 ± 6.4 | 27.2 ± 4.3 | −0.1 (−2.4–2.2) | 0.92 |

Notes.
*VD, vitamin D; 25(OH)D, 25-hydroxyvitamin D; AST, aspartate aminotransferase; ALT, alanine aminotransferase; TGF-$\beta$1, transforming growth factor beta 1; TIMP-1, tissue inhibitor of matrix metalloproteinase-1; MMP-9, matrix metalloproteinase 9; P3NP, amino terminal type III procollagen peptide.

**Table 4 Sensitivity analysis.** The median (25th–75th percentile) serum biochemical marker levels at baseline and 6 weeks, and the median change (95% CI) in serum biochemical marker levels from baseline to 6 weeks of follow-up, in the VD versus the placebo arm as a reference, from median quantile regression.

| Serum Marker | Week | VD Median (IQR) | Placebo Median (IQR) | Median difference (95% CI) in change from baseline to 6 weeks in VD vs placebo arm | p value |
|---|---|---|---|---|---|
| 25(OH)D level (ng/mL) | 0 | 19.2 (19.2–21.3) | 16.9 (14–19.1) | | |
| | 6 | 39.5 (36.7–46.5) | 17.6 (15.1–21.1) | 24.2 (20.8–27.6) | <0.001 |
| TGF-$\beta$1 level (ng/mL) | 0 | 14.8 (11.5–21.1) | 15.0 (9.2–17.8) | | |
| | 6 | 15.8 (10.1–22.5) | 15.6 (8.8–20.3) | −1.5 (−3.2–0.22) | 0.09 |
| TIMP-1 level (ng/mL) | 0 | 232.0 (200.7–275.1) | 226.4 (199.9–261.1) | | |
| | 6 | 229.0 (197.8–275.5) | 229.2 (189.0–276.0) | −13.2 (−33.2–6.7) | 0.19 |
| MMP-9 level (ng/mL) | 0 | 584.8 (345.2–802.1) | 449.4 (339.0–637.5) | | |
| | 6 | 615.1 (329.5–890.8) | 486.0 (363.7–721.7) | −9.2 (−146–127.7) | 0.89 |
| P3NP level (ng/mL) | 0 | 27.0 (26.1–31.0) | 28.1 (25.6–31.0) | | |
| | 6 | 25.8 (24.1–28.8) | 26.6 (23.8–29.7) | −0.1 (−2.8–2.5) | 0.92 |

Notes.
*VD, vitamin D; 25(OH)D, 25-hydroxyvitamin D; TGF-$\beta$1, transforming growth factor beta 1; TIMP-1, tissue inhibitor of matrix metalloproteinase-1; MMP-9, matrix metalloproteinase 9; P3NP: amino terminal type III procollagen peptide.

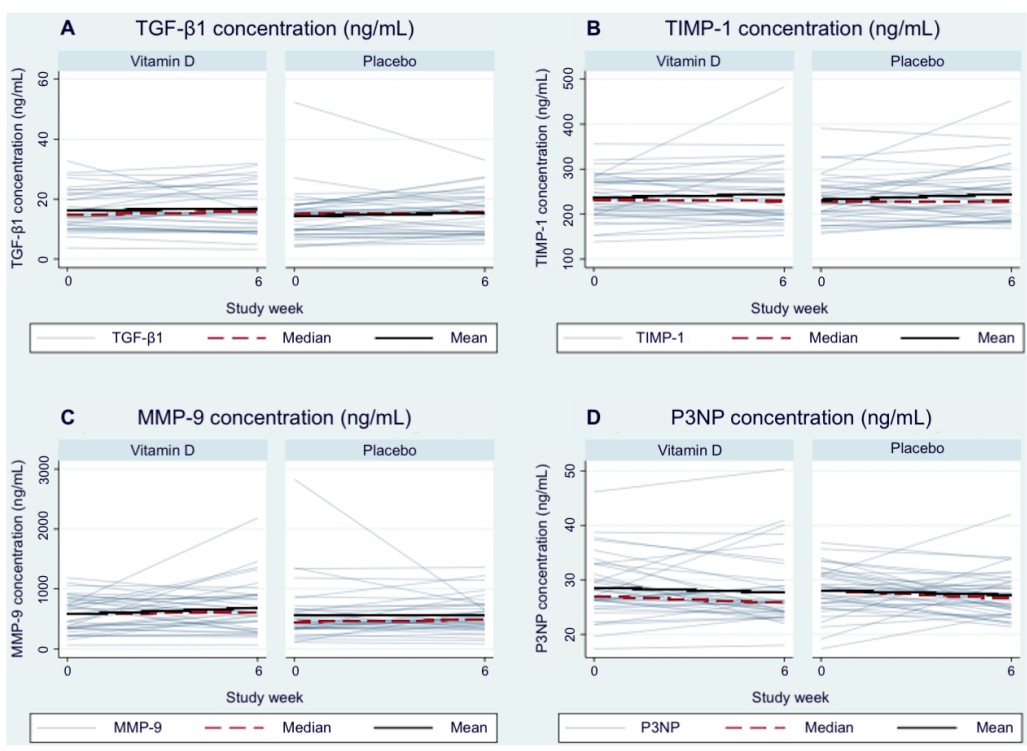

**Figure 3** Line plot showing the individual serum marker levels for each patient connected with the levels at 6 weeks by randomized treatment arm. The overall median and mean levels at weeks 0 and 6 are connected by a dashed and solid line, respectively. (A) TGF-$\beta$1. (B) TIMP-1. (C) MMP-9. (D) P3NP.

Mean P3NP levels slightly decreased from the baseline in both groups. There was no significant difference in the mean P3NP change score from baseline to week 6 between the VD and placebo groups ($-0.1$ ng/mL (95% CI [$-2.4$–$2.2$]), $p = 0.92$).

## Sensitivity analysis

We conducted quantile regression, using the median as another measure of the central tendency, to assess whether extreme outliers in the dataset had a strong influence on the observed associations. In both VD and placebo groups, the median, 25th and 75th percentile values of each marker were very similar at baseline and week 6 visits (Fig. 3, Table 4). The differences in the mean and median changes in marker concentrations from baseline to week 6 between randomized arms were of similar magnitude and direction, with the exception of MMP-9. For this latter marker, the mean change was greater, but the median change was smaller, in the VD group than in the placebo group. However, neither result was statistically significant, and the 95% CI around both the mean and median differences were wide.

## Subgroup analysis

A subgroup analysis was performed according to the severity of VD deficiency/insufficiency, which showed no significant changes in serum hepatic fibrogenesis markers in both groups.

Additional subgroup analyses based on sex, cirrhosis, liver fibrosis severity, and a 25(OH)D level <10 ng/ml at baseline also revealed similar results (Tables S1 and S2).

## DISCUSSION

In this randomized, double-blind, placebo-controlled study, 6-week VD supplementation in patients with CHC after HCV eradication by DAA did not help attenuate liver fibrosis as assessed by fibrogenesis markers. Additionally, the effect of VD supplementation did not vary according to baseline serum 25(OH)D levels and liver fibrosis severity.

Activation of HSCs to myofibroblast-like cells by TGF-$\beta$1 results in fibrillary component accumulation, which leads to progressive liver fibrosis (*Lee, Wallace & Friedman, 2015*). TGF-$\beta$1 also regulates TIMP-1 production, which subsequently inhibits the breakdown of excessive ECM (*Flisiak et al., 2005*). In contrast, a shift toward fibrolytic activity results in the reversal of liver fibrosis. MMPs function as the main enzymes responsible for fibrillary matrix cleavage (*Wynn, 2007*). P3NP is a breakdown peptide produced during the healing process, and is widely used as a marker for hepatic fibrolysis. A decline in P3NP levels correlates with an increase in cellular fibrosis degradation and indicates liver fibrosis regression (*Nielsen et al., 2015*). Based on their involvement in hepatic fibrogenesis, these cellular cytokines and enzymes have been used as markers for an indirect assessment of hepatic fibrogenesis (*Komolmit et al., 2017*; *Liu et al., 2012*)

Apart from its classical action on calcium metabolism, VD is involved in various cellular regulatory processes. Immunomodulatory, anti-inflammatory, and anti-fibrotic properties of VD have been reported in numerous studies. VD plays a crucial role in the functioning of the innate and adaptive immune systems (*Aranow, 2011*; *Holick, 2007*). Further, VD has also been demonstrated as an innate antiviral agent; it exerts an inhibitory effect on HCV replication in human cells (*Gal-Tanamy et al., 2011*). Regarding the association of VD and hepatic fibrogenesis, VD receptor (VDR) mRNA in HSCs is upregulated in response to TGF-$\beta$1 (*Beilfuss et al., 2015*). A study by Ding et al. showed spontaneous liver cirrhosis in *VDR* knockout mice, which was mediated by the TGF-$\beta$/VDR/SMAD signaling circuit (*Ding et al., 2013*). In addition, 1,25(OH)D stimulated VDR expression in primary rat HSCs and suppressed TIMP-1 mRNA expression, resulting in an improvement in fibrotic score in thioacetamide-induced liver fibrosis (*Abramovitch et al., 2011*). Furthermore, a synthetic VD analogue, calcipotriol, has been found to prevent liver cirrhosis in a mouse model (*Wahsh et al., 2016*). In a human study, Beilfuss et al. demonstrated the role of VD in liver fibrogenesis via the crosstalk between VD and the TGF-$\beta$1 signaling pathway. With VD supplementation, a reduced fibrogenic response in human HSCs was observed (*Beilfuss et al., 2015*). Based on the aforementioned evidence, at least three main properties of VD that could influence the fibrinolytic state can be postulated: anti-viral (i.e., suppression of viral replication), anti-inflammatory, and anti-fibrotic properties.

In our previous study, restoration of VD levels in patients with CHC resulted in dynamic shifts of serum fibrosis markers toward anti-fibrotic activity. The levels of pro-fibrotic cytokines (i.e., TGF-$\beta$1 and TIMP-1) significantly decreased in the VD supplementation group, whereas the levels of anti-fibrotic enzymes (i.e., MMP-2 and
MMP-9) were significantly higher in the VD supplementation group than in the placebo group (*Komolmit et al., 2017*). This evidence highlights and supports the protective role of VD in the hepatic fibrogenesis process. In our current study, we aimed to explore the effect of VD replacement after curative HCV therapy in order to ascertain whether VD mainly facilitates fibrolytic processes by halting fibrogenesis, not by its effect on HCV replication or inflammation. Based on our results, VD restoration after HCV eradication did not cause significant changes in serum markers associated with hepatic fibrogenesis. Hence, one could assume that VD, at the clinical level, might not help facilitate hepatic fibrolysis. In addition, the effect of VD on the improvement in fibrogenesis markers, as shown in our previous study, might not be directly achieved via the intrinsic fibrogenesis cascades, as observed in several experimental results. Thus, apart from HCV eradication and the resolution of liver inflammation, VD restoration after DAA treatment may not provide any further benefit in the reversal of hepatic fibrogenesis. This seems to suggest that VD might regulate and influence hepatic fibrogenesis through its immunomodulatory rather than its anti-fibrotic property. While VD might have a role in hepatic fibrosis regulation, its contribution is not strong enough to affect the associated serum markers. Investigation on changes in local tissue levels to elucidate this observation may be warranted.

The baseline TGF-$\beta$1 levels in the current cohort could be another possible explanation for the major difference in results between our previous and current studies. Unlike our previous study, the baseline TGF-$\beta$1 levels were similar to those observed in healthy volunteers (*Flisiak et al., 2005*; *Grainger, Mosedale & Metcalfe, 2000*), and were consistent with the levels in successfully treated patients with CHC in studies conducted in the interferon era (*Flisiak et al., 2005*; *Kotsiri et al., 2016*). Given the normalized baseline serum TGF-$\beta$1 levels, a large number of participants would be needed to detect differences in the outcome of interest. It is more likely that, at this stage in clinical trials, an investigation into the hepatic tissue expression levels of each marker would be appropriate to further test our hypothesis. However, in the era of DAA therapy for CHC, a paired liver biopsy has ethical implications.

In contrast to TGF-$\beta$1 levels, the MMP-9 and P3NP levels at baseline were higher than those in healthy population (*Latronico et al., 2016*; *Teare et al., 1993*). A closer analysis of the hepatic fibrolytic processes revealed a trend towards fibrosis degradation as evidenced by the changes in MMP-9 and P3NP levels; however, the changes did not reach statistical significance.

Apart from VD levels, genetic variation in the *VDR* gene might be one of the factors with an influence on the study findings. Since *VDR* polymorphism has been shown to be associated with liver fibrosis and potentially determines the VD response (*Baur et al., 2012*; *Estrabaud et al., 2012*), the negative results of the current study might be partly due to *VDR* genetic variation. However, this assumption cannot be confirmed in the current clinical setting.

To the best of our knowledge, this is the first study to determine the effect of VD on liver fibrosis in patients with CHC after DAA treatment. We presented some additional evidence on the influence of VD on hepatic fibrogenesis. Nevertheless, this study has some limitations. First, several factors, such as diet, sunlight exposure, and supplementary

products, could affect the measured serum VD levels. To minimize the possible influence of these factors and other potential biases, we employed a double-blind, placebo-controlled study design. Moreover, the participants were advised to maintain their usual activities of daily living and avoid any supplements during the study period. Second, histological evidence in the liver parenchyma would ideally be a gold standard to evaluate liver fibrosis. However, paired liver biopsy is an invasive procedure, and thus, is not free of risk. Owing to ethical and practical considerations, we decided to employ a non-invasive approach using surrogate serum markers to evaluate fibrogenesis. Lastly, we selected a 6-week time point because, in our previous study, VD levels were normalized in most of the patients by VD supplementation within this time period. As hepatic fibrogenesis is highly dynamic, and the half-lives of serum fibrogenesis cytokines and enzymes are relatively short, with spans of hours and days, we believe that a local immunological change would be initially demonstrated within this time period. However, since the reversal of hepatic fibrogenesis could take months or years to show a clinical benefit, a short follow-up period may be inadequate for detecting any significant difference. Hence, future long-term studies are needed to further our understanding.

## CONCLUSIONS

Short-term VD replacement after HCV eradication by DAA did not improve serum hepatic fibrogenesis marker levels and, thus, might not clinically facilitate the amelioration of residual liver fibrosis. Nevertheless, further investigation to determine whether a higher dose (VD derivative) for a longer period could potentially help ameliorate liver fibrosis is warranted.

## ACKNOWLEDGEMENTS

We would like to express our gratitude to the staff of the Division of Gastroenterology, Department of Medicine, Center of Excellence in Liver Diseases, King Chulalongkorn Memorial Hospital, and Center of Excellence in Clinical Virology, Faculty of Medicine, Chulalongkorn University for their technical assistance and clinical support.

### Funding

This work was supported by the Ratchadapiseksompotch Fund, Faculty of Medicine, Chulalongkorn University (RA59/074), the Ratchadaphiseksomphot Endowment Fund of hepatic fibrosis and cirrhosis research unit (GRU 6105530009-1), the Research Chair Grant from the National Science and Technology Development Agency (P-15-50004), and the Center of Excellence in Clinical Virology (GCE 59-009-30-005). The funders had no role in study design, data collection and analysis, decision to publish, or preparation of the manuscript.

## Grant Disclosures

The following grant information was disclosed by the authors:

Ratchadapiseksompotch Fund.

Faculty of Medicine, Chulalongkorn University: RA59/074.

Ratchadaphiseksomphot Endowment Fund of Hepatic Fibrosis and Cirrhosis Research Unit: GRU 6105530009-1.

Research Chair Grant from the National Science and Technology Development Agency: P-15-50004.

Center of Excellence in Clinical Virology: GCE 59-009-30-005.

## Competing Interests

The authors declare there are no competing interests.

## Author Contributions

- Supachaya Sriphoosanaphan and Piyawat Komolmit conceived and designed the experiments, performed the experiments, analyzed the data, prepared figures and/or tables, authored or reviewed drafts of the paper, and approved the final draft.
- Kessarin Thanapirom conceived and designed the experiments, prepared figures and/or tables, and approved the final draft.
- Stephen J. Kerr analyzed the data, prepared figures and/or tables, and approved the final draft.
- Sirinporn Suksawatamnuay performed the experiments, analyzed the data, prepared figures and/or tables, and approved the final draft.
- Panarat Thaimai, Sukanya Sittisomwong, Kanokwan Sonsiri and Nunthiya Srisoonthorn performed the experiments, prepared figures and/or tables, and approved the final draft.
- performed the experiments, prepared figures and/or tables, and approved the final draft.
- Nicha Teeratorn, Natthaporn Tanpowpong, Bundit Chaopathomkul, Sombat Treeprasertsuk and Yong Poovorawan conceived and designed the experiments, prepared figures and/or tables, and approved the final draft.

## Human Ethics

The following information was supplied relating to ethical approvals (i.e., approving body and any reference numbers):

This study was reviewed and approved by the Ethics Committee and Institutional Review Board (IRB) at Chulalongkorn University, Bangkok, Thailand and was performed in accordance with the Declaration of Helsinki (1989) of the World Medical Association (IRB number: 707/60).

## Clinical Trial Ethics

The following information was supplied relating to ethical approvals (i.e., approving body and any reference numbers):

This trial was registered with the Thai Clinical Trials Registry (TCTR) based on the World Health Organization criteria on 22 November 2017.

## Data Availability

Data are available at Dryad:

Komolmit, Piyawat (2020), Effect of vitamin D supplementation in patients with chronic hepatitis C after direct-acting antiviral treatment: a randomized, double-blind, placebo-controlled trial, Dryad, Dataset, https://doi.org/10.5061/dryad.573n5tb4h.

The codebook for categorical data is available in the Supplementary Files.

## Clinical Trial Registration

The following information was supplied regarding Clinical Trial registration:

TCTR20171206003.

## Supplemental Information

Supplemental information for this article can be found online at http://dx.doi.org/10.7717/peerj.10709#supplemental-information.

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
