# Peer review of "Effect of vitamin D supplementation in patients with chronic hepatitis C after direct-acting antiviral treatment: a randomized, double-blind, placebo-controlled trial"

_PeerJ, doi:10.7717/peerj.10709_

## Round 0.1 · original submission · Major Revisions

Thank you for submitting the manuscript to PeerJ for possible publication. I, Tuan Nguyen, was assigned to handle your your manucript which has now been reviewed by two experts in the field and their comments are attached for your perusal. Both reviewers appreciate the importance and relevance of your study, but they also raise some issues concerning the methodology that I hope you can address in your next submission.

As an Academic Editor, I have also gone through your data, and would like to invite you to comment on the following points:

1. Since this was a randomized controlled trial, did you register the trial with a recognized registry? Did you have a plan of analysis? Your "data analysis" in the appendix is not detailed enough; it is too generic.

2. The Abstract needs to be rewritten to provide information pertaining to the randomization, measurement of vitamin D, and duration of follow-up.

3. Could you comment on the intrasubject variability in vitamin D measurements by Liaison essay? I presume that you measured D3, not D2? This should be clarified.

4. The sample size calculation must be stated in more detail. What was the hypothesized difference? I consider that the sample size based on the mean change is flawed (if not to say wrong), because the change is likely to be dependent on baseline levels. It should be based on the DIFFERENCE in post-treatment levels between treatment groups. I share the concern of a reviewer on your sample size.

5. The analysis is flawed. Because each patient was measured multiple times, the proper analysis should be a mixed effects model, not t-test or Wilcoxon signed-rank test. I strongly suggest that you involve a professional biostatitician to TOTALLY re-analyze the data.

6. The presentation of data can be further improved. The boxplot is ok, but it is not informative. I want to see a line graph showing the vitamin D levels for EVERY participants stratified by treatment group (ie spaghetti plot). I also want to see the result of of the mixed effects analysis.

7. The conclusion that "VD supplementation in CHC patients after DAA treatment does not play an important role in hepatic fibrosis regression and may not expedite the residual liver fibrosis healing process" may be true, but I need to see more data to be convinced.

Other issues:

8. You must and should report EXACT p-value (eg p = 0.001), not relative P-values (eg "p < 0.01"). You should report 95% confidence interval of difference between two groups.

9. What is " Delta changes"? (line 253) Should it be simply "Changes"?

10. Please note that the standard abbreviation is "25(OH)D", not "25(OH)VD".

Reviewer 1 ·

Basic reporting

No comment

Experimental design

As comments below

Validity of the findings

As comments below

Additional comments

This study aimed to evaluate the effect of VitD supplementation on hepatic fibrosis in CHC after curative treatment with DAA. The surrogate markers in this study were fibrogenic and fibrolytic markers. This study is interesting but showed negative outcomes; however need to address the followings:
1. Due to patients with chronic liver disease including cirrhosis, 25 hydroxylage is defected and can not converse to active form (1,25(OH)2D). Is it qualify to measure 25(OH)D level. This is acceptable for general population but not in chronic liver disease or cirrhotic patients.
2. In healthy population, 25(OH)D at 24 ng/ml is accepted to be adequate except high risk subjects for osteoporosis or fracture. Why this study recruited patients who vitD level 30 ng/ml.?
3. VitD level at 6 wk after replacement and measure only 1 time can not reflect the results (hepatic fibrosis), how the author conclude that VitD supplement is not effective.
4. The author did not mention for baseline markers. Theses were normal or not? If in normal range at baseline, it can explain the negative results in this study. For example when compare the baseline markers (TGFB1, TIMP-1, MMP-3 levels) with previous author study in 2017, were very high in curative treatment with DAA.
5. This study measured only markers without liver biopsy findings, the conclusion in the study is beyond the current finding.
6. The classification of vitamin D status in Table was incorrect.
7. Table 4,5 and figure 2&3 repeated with previous table.
8. How about the effect of supplementation in VitD level <10 group?

Reviewer 2 ·

Basic reporting

No comment

Experimental design

1. It is not clear to me whether the sample size was determined. The assumption was that VD may facilitate further fibrosis amelioration which is regulated by 2 activities: fibrogenic (TGF-β1, TIMP-1) and fibrolytic (MMP-9, P3NP). But the authors calculated sample size only based on the mean changes in serum TGF-β1 level for fibrogenic. Furthermore, there was subgroup analysis according to VD level. If so, these issues should be considered in assumption to calculate sample size for this study.
2. The reduced liver function, not decompensated liver cirrhosis, can impact on the hydroxylation of VD into 25(OH)VD. The authors need to concern this issue in excusive criteria.
3. Please clarify sensitiviy and specitivity of biomarker tests for liver fibrogenesis and fibrolytic.
4. The rationale for choosing the time intervention 6 weeks is not clear. This is only the required time for supplement VD, not for immunological change of fibrosis.

Validity of the findings

1. Table 5 should be reformed with accurate p-value, not only all p-value >0.05.
2. I consider that the conclusion, "VD supplementation in CHC patients after DAA treatment does not play an important role in hepatic fibrosis regression… " is not supported by the data. The enough time for evaluating immunological change of fibrosis should be assessed before the authors could reach that conclusion.

Additional comments

This study attempts to address a clinically important issue, but I am not quite convinced that the authors have really addressed their aims. The method is described with insufficient detail and information to replicate.

---

## Round 0.2 · accepted · Accept

Thank you for your response to reviewers' and my comments. We are happy that you have addressed all concerns.

Reviewer 1 ·

Basic reporting

no comment

Experimental design

no comment

Validity of the findings

no comment

Additional comments

no further comment

Reviewer 2 ·

Basic reporting

No comment

Experimental design

No comment

Validity of the findings

No comment

Additional comments

All comments addressed sufficiently. No more questions.